# Reinforcement Learning Enhanced Full-Duplex Spoken Dialogue Language Models for Conversational Interactions

**Chen Chen**    **Kevin Hu**    **Chao-Han Huck Yang**    **Ankita Pasad**
**Edresson Casanova**    **Weiqing Wang**    **Szu-Wei Fu**    **Jason Li**
**Zhehuai Chen**    **Jagadeesh Balam**    **Boris Ginsburg**
NVIDIA

## Abstract

Mainstream spoken dialogue language models (SDLMs) primarily handle turn-based interactions by alternating between processing user speech and generating responses. Recently emerging full-duplex SDLMs have showcased more natural and engaging conversational performance by simultaneously listening and speaking. However, the complex dynamics of human conversation introduce unique challenges to full-duplex SDLMs: Beyond generating reasonable responses, these models must exhibit diverse and prompt conversational behaviors in real-time interactions with the user. In this work, we present an efficient full-duplex SDLM optimized by **O**nline **R**einforcement with **I**nteractive **S**peech **E**valuation (ORISE). In ORISE, we design a customized reward function derived from automated annotations of online generated speech to guide the model toward well-formed and speech-text aligned responses. Experimental results show that ORISE effectively improves robustness and accuracy in handling conversational dynamics, including turn-taking, user barge-in, and backchanneling. Furthermore, ORISE enables the model to adapt to unseen noise conditions without relying on any labeled data, demonstrating the generalization of ORISE in real-world scenarios.

## 1 Introduction

Spoken dialogue is the most intuitive form of human-computer interaction, enabling users to communicate effortlessly and naturally with AI agents (Rebman Jr et al., 2003). Consequently, research on spoken dialogue language models (SDLMs) has emerged as an important topic within conversational AI (Freed, 2021), exemplified by popular voice assistants such as Siri and Alexa, which facilitate various aspects of daily life.

As shown in Figure 1 (a), most existing SDLMs rely on turn-based interactions and process user speech inputs one segment at a time to generate speech or textual responses (Kulkarni et al., 2019). However, these half-duplex systems, which must alternate strictly between "listening" and "speaking" modes, struggle to handle complex conversational dynamics commonly observed in human dialogue, such as user barge-in or backchanneling (Lin et al., 2025a). Therefore, developing full-duplex SDLMs has emerged as a more attractive goal (Wang et al., 2024b) to simultaneously process streaming user inputs and generate agent voices, as shown in Figure 1 (b). This simultaneous modeling of input and output channels facilitates a more responsive and flexible form of interaction (Défossez et al., 2024).

Beyond comprehending speech content, full-duplex SDLMs are expected to learn timing information in spoken conversations to provide prompt responses. As illustrated in Figure 1 (b), we highlight three key conversational behaviors: (1) **User barge-in**: agent should gracefully allow users to interrupt its ongoing speech and switch seamlessly to listening mode; (2) **Turn-taking**: the agent should determine appropriate moments to begin speaking once the user finishes the utterance; and (3) **Backchanneling**: the agent should remain unaffected by users' short affirmative response (e.g., "yeah", and "Mm-hmm", etc.) or third-party voice. Additionally, considering the variability of conversational environments,

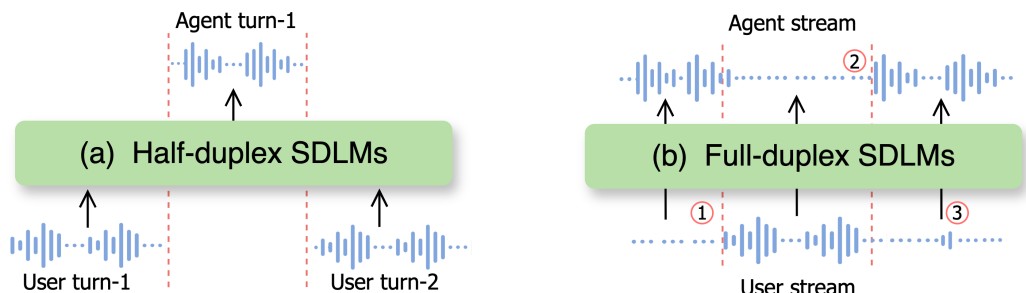

Figure 1: (a) Half-duplex SDLMs with turn-based interaction. (b) Full-duplex SDLMs can listen and speak simultaneously, which allows the system to handle conversation dynamics, including ① User barge-in, ② turn-taking, and ③ user backchanneling.

the overall system needs to demonstrate robustness against background noise. Collectively, these factors significantly increase the complexity of conversation modeling and pose challenges for full-duplex SDLMs.

To address these challenges, this work aims to develop a robust SDLM to effectively handle real-time conversational dynamics. We first present an efficient and direct speech-to-speech modeling approach for adapting a text-based LLM into a full-duplex SDLM. Building on this model, we focus on applying reinforcement learning (RL) as a post-training optimization to refine the agent's conversational behaviors. This motivation stems from the long-standing utilization of RL to mitigate the mismatch between training and evaluation in sequence generation problems (Bahdanau et al., 2016; Rennie et al., 2017; Prabhavalkar et al., 2018), a.k.a., *exposure bias* introduced by the Teacher-forcing algorithm (He et al., 2019b).

We propose ORISE, which adopts an online sampling and evaluation strategy that leverages a voice activity detection (VAD) model (Team, 2021) and automatically assigns multiple rewards for the predicted speech. Through policy gradient optimization, the likelihood of sequences exhibiting desirable conversational behaviors is promoted, allowing the model to pursue higher reward expectation. Experimental evidence demonstrates that ORISE effectively improves the robustness in handling conversational behaviors, including turn-taking, user barge-in, and backchanneling. Furthermore, as our rule-based reward function does not requires labeled examples, we apply ORISE to unseen background noise and improve the accuracy of barge-in and backchanneling in a fully unsupervised adaptation manner. In summary, our contribution is as follows:

- We propose ORISE, the first RL-based optimization method specifically designed for full-duplex SDLMs. ORISE leverages automated annotations to guide the model's conversational behaviors using self-generated samples, effectively mitigating the exposure bias commonly observed in long-sequence generation under SFT.

- Enhanced by ORISE, our SDLM is more robust and efficient, achieving higher accuracy and lower latency in handling conversational dynamics such as user barge-in and backchanneling. Our SDLM delivers high-quality responses with only 1/6 the parameter size of Moshi (Défossez et al., 2024).

- ORISE is also empirically shown to mitigate the model performance degradation in unseen background noises **without** relying on any supervised labels, demonstrating its practical value for deploying conversational AI in real-world scenarios.

## 2 Related Work

### 2.1 Full-Duplex Spoken Dialogue Language Models

To address the limitation of half-duplex models, recent works such as GPT-4o voice mode (Hurst et al., 2024) have been proposed to build full-duplex models capable of simultaneous speaking and listening capacities (Lin et al., 2025b). According to modeling

approaches, existing full-duplex models can be categorized into two types: cascaded models and end-to-end models.

Cascaded approaches break down the process into distinct stages—first converting speech to text and then handling the linguistic content. For example, Wang et al. (2024a) propose using control, speak, and listen tokens on a large language model with additional perception and motor modules, although this approach still depends on separate speech-to-text and text-to-speech components. Similarly, MiniCPM-Duplex (Zhang et al., 2024c) and MiniCPM-Duo (Xu et al., 2024) incorporate time-sliced text tokens via time-synchronous methods, while systems like VITA (Fu et al., 2024) and Freeze-Omni (Wang et al., 2024c) accept raw speech inputs to speed up inference, though they still require text to complete the loop.

In contrast, end-to-end two-channel models aim to jointly process both user and agent channels without intermediate conversion. For instance, dGSLM (Nguyen et al., 2023) employs a Siamese network with cross-attention to handle two-channel dialogue, and Moshi (Défossez et al., 2024) utilizes parallel streams to separately process speaker and listener speech. SyncLLM (Veluri et al., 2024) introduces a time-synchronous mechanism for autoregressive decoding across channels, while OmniFlatten (Zhang et al., 2024b) adopts progressive post-training to jointly process flattened sequences of speech and text tokens. Other works, including SALMONN-omni (Yu et al., 2024), MinMo (Chen et al., 2025), Parrot (Wang et al., 2025), RTTL-DG (Mai & Carson-Berndsen, 2025), and neural FSM in (Wang et al., 2024b) explore techniques such as augmenting state tokens, next-token-pair prediction, and dialogue management to refine two-channel processing.

## 2.2 Reinforcement Learning in Language Generation

Before the advent of LLMs, optimization methods based on reinforcement learning had already been widely applied to language generation tasks (He et al., 2019a;b). For example, in the speech domain for ASR tasks (Prabhavalkar et al., 2018), image domain for image captioning (Rennie et al., 2017), and NLP for machine translation (Bahdanau et al., 2016), these methods aim to optimized SFT models based on the traditional REINFORCE algorithm (Williams, 1992) or improved variants (Zhang et al., 2021).

In the LLM era, RL-based optimization methods are increasingly explored in the form of RLHF technique (Ouyang et al., 2022). Given that LLMs already exhibit strong generation capabilities, RLHF primarily serves as a post-processing technique to align model outputs with human preferences in various scenarios (Dai et al., 2023; Dong et al., 2024). Early RLHF algorithms rely on an independent reward model to score generated samples (Zheng et al., 2023), whereas later approaches adopt implicit reward modeling, directly learning human preferences from chosen/rejected pairs (Rafailov et al., 2024; Meng et al., 2024).

## 3 Speech to speech (S2S) Full-Duplex Modeling

An overview of our proposed full-duplex SDLM is illustrated in Figure 3. Given a (possibly noisy) user stream $X$, our duplex S2S model $\theta$ is designed to concurrently process the incoming signal while streaming predictions for both text $Y^1$ and speech $Y^2$ outputs. This simultaneous "listening" and "speaking" autoregressive generation process is formalized as follows:

$$P_\theta(Y^1, Y^2 \mid X) = \prod_{t=1}^{T} P\left[\langle y_t^1, y_t^2 \rangle \mid y_{<t}^1, y_{<t}^2, X_{\leq t}\right], \tag{1}$$

where $X_{\leq t}$ represents the user stream up to time step $t$, and $y_t^1$ and $y_t^2$ denote the text and speech outputs at time step $t$, respectively. We hereby highlight following key factors of modeling a text-only LLM to a full-duplex SDLM:

**Speech Tokenization.** We employ the neural audio codec to represent the speech predicted by SDLMs, i.e., the agent stream can be represented as an acoustic matrix with length $T$ and $K$ channels of codebooks. Unlike commonly used RVQ in TTS technique (Wang et al., 2023), we adopt NanoCodec (Anonymous, 2025) with Finite Scalar Quantization (Casanova et al., 2025) to ensure independence among multiple codebooks. This independence allows

all $K$ codebooks to be predicted in parallel at each timestep, thereby enabling fully parallel modeling with low latency.

**Alignment between generated text and speech.** Due to different modalities, $Y^2$ is typically longer than $Y^1$, making synchronous prediction unfeasible. To align $Y^1$ and $Y^2$, we introduce a padding mechanism for the text channel to coordinate a bi-modal response. As illustrated in the Figure 3, after the text output is predicted, the model continues to generate a special $\langle pad \rangle$ token until the predicted audio codecs reach the end of sequence. This approach eliminates the need for word-level alignment like Moshi (Défossez et al., 2024), and even if the prediction process of $Y^2$ is interrupted by the user, the generated $Y^1$ can serve as context for subsequent dialogue.

**User stream perception.** We employ a pretrained streaming speech encoder to extract continuous embeddings $X' \in \mathbb{R}^{T \times d_1}$ from the user channel in real time. The frame rate of the encoder is set to match the frame rate of the audio codec model. Consequently, for each time step $t$, a projection layer maps $X'_t$ to match the embedding dimension of LLM, $d_2$ after which $(y^1_{t-1}, y^2_{t-1}, X'_t)$ are fused at the embedding level as the input to the LLM. Notably, compared to discretizing user speech using an audio codec (Défossez et al., 2024), the streaming speech encoder can be trained end-to-end, capturing subtle nuances in the user stream and more flexibly adapting to diverse acoustic environments.

**LLM backbone.** To adapt any text-based LLM to receive and predict audio tokens, we first expand its original word embedding matrix $W \in \mathbb{R}^{v_1 \times d_2}$ with $K$ layers of codebooks, resulting in a new word embedding matrix $W' \in \mathbb{R}^{(v_1 + v_2 \times K) \times d_2}$. After obtaining the hidden representation $h_t$ from the final transformer block, the original LLM's linear layer is reused to map $h_t$ to the logits of vocabulary size $v_1$, while $K$ parallel linear layers independently map $h_t$ to the logits with size $v_2$ for each audio codec codebook. In practice, we observe that the text channel typically learns knowledge first and predominantly influences conversational behavior. Therefore, two techniques are implemented to enhance model performance. First, the text channel is assigned a higher weight when computing the cross-entropy loss. Second, a one-token delay (i.e., 80ms) is introduced between the text channel and the audio channel. Our SDLM is trained with cross-entropy loss as follows:

$$\mathcal{L}_{CE} = -\sum_{t=1}^{T} \left[ \lambda_1 \log p \left( y^1_t \mid \mathbf{h}_t \right) + \lambda_2 \log p \left( y^2_{t-1} \mid \mathbf{h}_t \right) \right] \tag{2}$$

where $\lambda_1$ and $\lambda_2$ are the hyper-parameters to balance the weights between speech and text channels.

**S2S Training Data.** Existing publicly available conversational datasets, such as Fisher (Cieri et al., 2004), capture human dialogue behaviors but are insufficient for training an helpful agent to effectively respond to a wide range of inquiries. To address this gap, we construct dialogue-style data featuring two speech streams: one for the user and one for the agent. When the user begins speaking, the agent is expected to predict silence; once the agent takes over the turn, appropriate silence is inserted into the user stream to allow the agent to respond. Further details are provided in the Appendix A.1.

## 4 Online Reinforce with Interactive Speech Evaluation (ORISE)

Given an SFT model, reinforcement learning aims to improve its generation quality by maximizing a customized reward $R$. In the mainstream RLHF, $R$ is typically determined by an independent reward model trained with human feedback, or directly learns human preference from annotated chosen/rejected pairs. To circumvent the high cost associated with human annotations, this paper focuses on **automated metrics** $\mathcal{R}$ to measure the conversational performance of generated samples, thus eliminating the need for any ground truth involvement throughout the optimization process. In general, the maximizing objective can be written as:

$$\mathbb{E}_{X \in \mathcal{D}, Y^1, Y^2 \in \pi_\theta}[\mathcal{R}(X, Y^1, Y^2)] - \beta \mathbb{D}_{KL}(\pi_\theta(\langle Y^1, Y^2 \rangle | X) \| \pi_{\text{ref}}(\langle Y^1, Y^2 \rangle | X)), \tag{3}$$

Table 1: Ideal agent behaviors in respond to different user speech intents, where 0 and 1 in transition denote "silence" and "speaking" modes of agent, respectively.

| Index | User intent | Transition | Agent Behavior Description | Reward |
|---|---|---|---|---|
| $\mathcal{C}_{2\text{-}1}$ | normal query | $0 \to 1$ | take the turn within delay $\Delta t_1$ | - |
| $\mathcal{C}_{2\text{-}2}$ | barge-in | $1 \to 0$ | stop speaking within delay $\Delta t_2$ | 1 |
| $\mathcal{C}_{2\text{-}3}$ | backchanneling | $1 \to 1$ | continue speaking until $t' + \Delta t_3$ | 1 |

where a reference model $\pi_{\text{ref}}$ with KL divergence penalty is introduced to prevent the model $\pi_\theta$ from making radical update, and $\beta$ is a balancing weight. To annotate the quality of generation, we define three automatic reward criteria for evaluating generated samples:

- $\mathcal{C}_1$: Turns Consistency. In multi-turn dialogues, the number of turns predicted by the agent should be equal to the number of user turns, excluding those backchannel responses: $\mathcal{R}_1 = -|\text{Number of turns}(X) - \text{Number of turns}(Y^2)|$, where the Number of turns($Y^2$) is examined by a VAD model Silero (2024).

- $\mathcal{C}_2$: Conversational behavior. Given a segment of user speech $X'$ with ending timestamp of $t'$, we define speaking state of agent as 1 and silence state as 0, and the desirable behaviors of agent are illustrated in Table 1. Due to the awareness of $t'$ and transition of user speech, the generated agent speech $Y^2$ can be detected by a VAD model to determine whether the desirable action has been executed, without any reliance on ground-truth. Since $\mathcal{C}_{2\text{-}1}$ is easy to learn from SFT, we define the $\mathcal{R}_2 = \mathbb{I}[\mathcal{C}_{2\text{-}2}] + \mathbb{I}[\mathcal{C}_{2\text{-}3}]$, where the indicator function $\mathbb{I}$ returns 1 if the condition inside is satisfied for the whole conversation, and 0 otherwise.

- $\mathcal{C}_3$: Alignment between $Y^1$ and $Y^2$. Since the agent generates text and speech independently, their content should remain consistent. To evaluate this, we employ an ASR model (Galvez et al., 2024) to transcribe the generated speech into text and then compute the word error rate (WER) between the predicted transcription and the generated text. Note that $\mathcal{R}_3$ is applied only to those samples that were not interrupted, as when a user barges in, the text channel, $Y^1$ may have more content than the speech channel, $Y^2$. In such cases, a high WER does not necessarily indicate a mis-alignment between the two predicted modalities.

To avoid introducing excessive hyperparameters, we directly sum the $\mathcal{R}_1 \sim \mathcal{R}_3$ to measure the quality of $Y^2$. Additionally, preference-based optimization can also be conducted in terms of positive/negative samples selected from these criteria. We consider preference optimization methods as baseline and introduce more details in the Appendix A.2.

For optimization, we adapt typical REINFORCE algorithm (Williams, 1992) with online sampling strategy. Specifically, based on user speech $X$, we perform online decoding to sample $N$ pairs of $\langle \hat{Y}^1, \hat{Y}^2 \rangle$ using $\pi_\theta$. Then the expectation in Eq. (3) is approximated using an empirical average over these samples, and the optimization objective is written as:

$$\mathcal{L} = -\frac{1}{N} \sum_{n=1}^{N} \left[ \left( \lambda_1 P_\theta(\hat{Y}_n^1 \mid X) + \lambda_2 P_\theta(\hat{Y}_n^2 \mid X) \right) \mathcal{A}_n - \beta \mathbb{D}_{\text{KL}}(\pi_\theta \| \pi_{\text{ref}}) \right], \qquad (4)$$

where $\lambda_1$ and $\lambda_2$ are keep consistent with SFT process in Eq. (2). $P_\theta(\hat{Y}_n^1 \mid X)$ and $P_\theta(\hat{Y}_n^2 \mid X)$ $\lambda_1$ are normalized by the sequence length T. $\mathcal{A}$ is the advantage function (Schulman et al., 2017), and KL-item is estimated using the approximator introduced by (Schulman, 2020):

$$\mathcal{A}_n = \frac{\mathcal{R}(X, \hat{Y}_n^2) - \text{mean}(\mathcal{R}(X, \hat{Y}^2))}{\text{std}(\mathcal{R}(X, \hat{Y}^2))}, \qquad (5)$$

$$\mathcal{R} = \mathcal{R}_1 + \mathcal{R}_2 + \mathcal{R}_3, \qquad (6)$$

$$\mathbb{D}_{\text{KL}}[\pi_\theta \| \pi_{\text{ref}}] = \sum_{t=1}^{T} \left[ \frac{\pi_{\text{ref}}\left(y_t^1, y_t^2 \mid X, y_{<t}^1, y_{<t}^2\right)}{\pi_\theta\left(y_t^1, y_t^2 \mid X, y_{<t}^1, y_{<t}^2\right)} - \log \frac{\pi_{\text{ref}}\left(y_t^1, y_t^2 \mid X, y_{<t}^1, y_{<t}^2\right)}{\pi_\theta\left(y_t^1, y_t^2 \mid X, y_{<t}^1, y_{<t}^2\right)} - 1 \right] \qquad (7)$$

In Eq. (4), if $Y_n^2$ obtain higher rewards than other samples, its likelihood generated by $\pi_\theta$ would be increase due to the positive value of $\mathcal{A}_n$, and vice versa. Additionally, although the reward is computed based on $Y^2$, the behavior of $Y^1$ remains largely consistent with it, as text channel is predicted one token ahead of the speech channels. The pseudo code of ORISE is summarized as Algorithms 1.

---

**Algorithm 1** ORISE Algorithm

---

**Require:** SFT model $\pi_\theta$, VAD model $\phi$, Unlabeled dataset $\mathcal{D} = \{X_i, t_i'\}_{i=1}^I$
**Ensure:** Optimized Model $\pi_\theta$
 1: Duplicate $\pi_\theta$ to get reference model $\pi_{\text{ref}}$
 2: **for** each training step **do**
 3:     Sample user input $X$
 4:     Obtain $N$ pairs of $\langle Y^1, Y^2 \rangle$ online samples with $\pi_\theta$
 5:     Calculate reward $R_n$ for $Y_n^2$ using $\phi$, $t'$, and states of $X$
 6:     Compute loss $\mathcal{L}$ in Eq. (4)
 7:     Update parameters: $\theta \leftarrow \theta - \eta \nabla_\theta \mathcal{L}$
 8: **end for**
 9: **return** Optimized model parameters $\theta$

---

## 5 Experimental Setup

### 5.1 Dataset

**Training Data.** To generalize spoken QA capability, we use a multi-speaker TTS model (Hussain et al., 2025) to synthesize context, question, and answer from text-based datasets such as *Alpaca* (Taori et al., 2023) and *MS-MARCO* (Bajaj et al., 2018). To prevent the model from overfitting to synthetic speech, we follow the approach proposed in (Noroozi et al., 2024) to construct an additional QA dataset using the Mistral LLM, where the user speech is a mixture of real and TTS-generated audio, referred to as the ASR-QA set. Furthermore, we construct two multi-turn dialogue datasets, *Topic* and *UltraChat*, to enhance the agent's capabilities to leverage context information. In *UltraChat*, the original text data (Ding et al., 2023) is reformulated into a dialogue format, while Topic simulates user-agent conversations based on a specified topic. For RL, only the *UltraChat* user speech is employed for optimization. Detailed data statistics is shown in Appendix A.1 Table 6.

**Background Noise.** We utilize WHAM (Wichern et al., 2019) to create noisy dataset, which are dynamically mixed into the user channel during training in an online manner. The signal-to-noise ratio (SNR) is randomly sampled from a range of 10dB to 30dB. For unseen noise setting, we randomly sample some noise from MUSAN (Snyder et al., 2015).

**User Backchannel.** To simulate user backchanneling, we first use GPT-4o to generate 20 affirmative short utterance that commonly occurs in dialogue backchanneling. We then sample 200 speakers from the LibriTTS dataset and randomly paired them with the generated phrases. Using multi-speaker TTS model (Du et al., 2024), we have synthesized a total of 4,000 distinct backchannel audio samples. Next, these backchannel samples are randomly selected during training and inserted into silent segments of the user channel that exceed 4 seconds. The insertion point is set to 2 seconds after the agent begins speaking to simulate real backchannel. During inference, since the agent's response time is uncertain, we ensure that the user backchannel is inserted before the agent completes its response.

### 5.2 Training Details and Baselines

Our model is implemented in PyTorch using the NeMo Toolkit (Kuchaiev et al., 2019) and trained on 32 A100 (80G) GPUs, with each GPU handling a batch duration of 1000 seconds. The speech encoder is derived from a 100M-parameter streaming pretrained model with an 80ms right context (NVIDIA, 2023), while the LLM is initialized from the 1.1B TinyLlama

| Model | E2E | Turn-Taking Latency (s) | Barge-in | | Backchannel Acc (%) | MOS |
|---|---|---|---|---|---|---|
| | | | Acc (%) | Latency (s) | | |
| Freeze-Omni | ✗ | 1.17* | 79.5* | 1.20* | - | **4.3** |
| dGSLM | ✓ | 0.57 | 85 | 0.86 | - | 2.2 |
| Moshi | ✓ | n.a. | 55.1 | 0.81 | - | 3.9 |
| ORISE | ✓ | **0.43** | **96.8** | **0.61** | **95.7** | 4.2 |

Table 2: Conversational Behaviors of different models under various conditions. Turn-taking and Backchannel are evaluated on *UltraChat*, and Barge-in is evaluated on *Impatient*. The number with "*" denotes it is reported in (Lin et al., 2025a), and "−" indicates that the model does not support the corresponding functionality.

(Zhang et al., 2024a). For tokenization, we employ a 32k SentencePiece model for text and a customized 0.6 kbps NanoCodec (Anonymous, 2025) for speech. The speech representation utilizes four codebook layers, each containing a vocabulary of 4,037 entries. The training loss is balanced across modalities, with weights set to $\lambda_1 = 3$ for text and $\lambda_2 = 1$ for speech. Optimization is performed using FusedAdam with an inverse square root annealing learning rate schedule, beginning with a peak learning rate of 3e-4 for SFT and 1e-5 for RL. To maintain stability, gradient clipping is enforced at a threshold of 1.0. The $\beta$ to balance the KL-item is set as 0.2.

We adopt two preference optimization baselines DPO (Rafailov et al., 2024) and IPO (Azar et al., 2024), as well as a baseline that does not rely on pairwise preferences, KTO (Ethayarajh et al., 2024). Since these methods are originally designed for LLMs, several necessary adaptations are made to apply them to SDLMs. Further implementation details are provided in the Appendix A.2.

### 5.3 Evaluation Dataset and Metric

**Response Quality.** We evaluate our model in two scenarios: 1) multi-turn conversations: *UltraChat*, *Roleplay*, and *Topic*, and 2) spoken QA reasoning: *ASR-QA* and *Alpaca*. All these test sets are unseen during training. To measure the response quality, we utilize **GPT score**[1] ranging from 0 to 10 based on the hypotheses and references of all the agent turns. The agent speech response is transcribed by ASR model[2] as GPT input. We also report the **UTMOS** automatically evaluate by pre-trained UTMOS model (Saeki et al., 2022). As a S2S model, we do not report the quality of the predicted text, while it is generally better than the predicted speech content in terms of quality and coherence.

**User Barge-in.** Since the response time of agent is unpredictable, the natural occurrence probability of user barge-in is relatively low. To evaluate the barge-in performance, we create a test set based on *ASR-QA*, called *impatient*, where the agent is given barely an average of 2 seconds to respond. Then the user barge-in occurs in more than 95% cases, showcasing a obvious mismatch with training set. The $\Delta t_1$ is set as 1.5 second, meaning that if the agent does not stop speaking within 1.5 seconds, it is considered a barge-in failure. We report both **accuracy** and average **latency** as metric to measure the barge-in performance.

**User Backchanneling.** Different with training, we insert random backchannel utterance in every response of agent turn. *UltraChat* is selected as test set due to its long agent response, and if agent stops speaking within 1.5 second, it is considered backchannel failure. We report the average **accuracy** calculated across the all backchannels.

**Turn-taking.** As the basic behavior for the model, we evaluate it by measuring the **latency** between the end of the user's speech and the start of the agent's response. Notably, this we introduce a 0.64-second delay in the training label to make the conversation more natural.

---

[1]https://openai.com/index/gpt-4o-mini-advancing-cost-efficient-intelligence/
[2]https://huggingface.co/nvidia/parakeet-tdt_ctc-110m

| ID | Models | LLM Backbone | Multiturn Conversation | | | Spoken QA | |
|----|--------|--------------|------------|----------|-------|--------|--------|
| | | | *UltraChat* | *Roleplay* | *Topic* | *ASR-QA* | *Alpaca* |
| 1 | Moshi | Helium-7B | 3.4 | 1.7 | 2.8 | 1.9 | 1.7 |
| 2 | SFT Model | Qwen2-1.5B | 3.7 | 3.7 | **6.1** | 6.3 | **3.3** |
| 3 | SFT Model | TinyLLaMA-1.2B | 3.5 | **4.6** | **6.1** | **7.8** | 2.9 |
| 4 | + ORISE | | **4.0** | 4.1 | 5.8 | 7.7 | 2.9 |
| 5 | GT+LLM | TinyLLaMA-1.2B | 6.4 | 4.9 | 5.5 | 5.8 | 5.0 |

Table 3: GPT-score of multi-turn conversation and spoken QA. GT+LLM denotes an oracle cascaded system which feeds every ground-truth user turn to the LLM.

| Metric | Test set | SFT Augmented by | | | RL Optimization | | | |
|--------|----------|------|------|------|------|------|------|-------|
| | | 0 % | 20% | 80% | DPO | IPO | KTO | ORISE |
| BLEU | | 10.5 | 10.0 | 10.1 | 6.2 | 8.3 | 5.5 | 9.8 |
| Barge-in-Acc | *UltraChat* | 83.0 | 74.1 | 70.7 | 66.9 | 70.9 | 75.5 | **83.7** |
| Backchannel-Acc | | - | 85.8 | 86.4 | 88.0 | 92.7 | **96.0** | 95.7 |
| Barge-in-Acc | *Impatient* | 94.5 | 94.1 | 96.5 | 94.7 | 94.7 | 95.2 | **96.8** |

Table 4: Comparison of conversational behavior improvements using RL, where "Augmented by 80%" is the starting point of RL optimization.

## 6 Result and Analysis

### 6.1 Main Results

We first report the conversational behavior in Table 2, where Moshi (Défossez et al., 2024), Freeze-Omni (Wang et al., 2024c), and dGSLM (Nguyen et al., 2023) are employed as baselines. We observe that: (1) For turn-taking, end-to-end models exhibit lower latency compared with cascaded model. The "n.a." entries for Moshi are due to its frequent behavior of responding before the user finishes speaking. In contrast, our model introduces a 0.64-second delay during training, while during inference, the latency dynamically varies between 0.4 and 0.9 seconds across different datasets. (2) For barge-in, our model demonstrates strong robustness compared with other baselines. Under both clean and noisy conditions, it significantly outperforms the baselines in terms of both accuracy and latency. (3) For backchanneling, our model achieves high accuracy, indicating its effectiveness in distinguishing normal user query and affirmative response.

In Table 3, we report our model's reasoning capabilities by comparing it against E2E baseline Moshi and an oracle cascaded system, where ground-truth transcriptions of user turns are directly provided to a language model (denoted as GT+LLM in Table 3). The dGSLM is not included in the comparison as it cannot function as an agent capable of responding to user requests. By comparing System ID-2 and System ID-3, we demonstrate that our S2S modeling can efficiently adapt different text-based LLM into a full-duplex SDLM. Furthermore, the comparison between System ID-3 and System ID-4 shows that ORISE effectively enhances dialog capabilities without compromising the model's reasoning performance. We attach listening examples are in anonymous link[3].

### 6.2 Effect of ORISE

We then evaluate the efficacy of RL-based optimization and report the accuracy in Table 4. We use the SFT model augmented by a certain percentage of synthesized user backchannel speech inserted into the training data. When it is 0%, any user voice is considered as barge-in for agent. Building upon the model trained with 80% backchannel augmentation,

---

[3]https://2025anonymous1.github.io/COLM-demo/

we apply various post-training optimization methods, including DPO, IPO, KTO, and our proposed ORISE. Additionally, since the reward function focuses solely on conversational behaviors, we report the BLEU score on the text channel to assess the model's ability to generate high-quality response content and to monitor whether the quality of the generated responses degrades due to the conservation-specific optimization.

From Table 4, we observe that post-training methods consistently improve backchannel accuracy, as backchannel events occur frequently and the model can simply continue speaking to receive a high reward. The $-\mathcal{R}_1$ and $\mathcal{R}_2$ (backchanneling) is visualized in Figure 2. However, for barge-in, different methods exhibit distinct behaviors across the *UltraChat* and *Impatient* evaluation settings. Despite the limited response window in *Impatient*, the agent demonstrates strong robustness and is often successfully interrupted across different models. In contrast, in the multi-turn conversations of *UltraChat*, the success rate of barge-ins drops—particularly as the proportion of backchannel examples increases in the training data. Compared to preference-based optimization methods, our approach offers two key advantages. First, it achieves balanced improvements in both backchannel and barge-in handling. Second, it demonstrates higher BLEU score, indicating that the quality of generated responses remains stable.

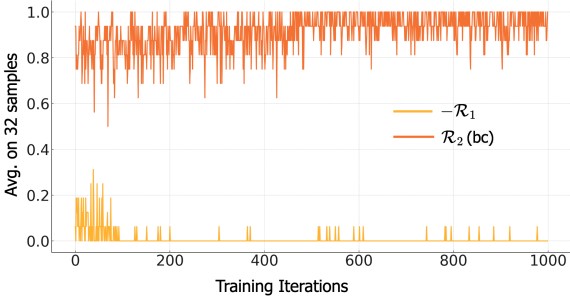

| Noise | Model | Accuracy | | |
|---|---|---|---|---|
| | | TT | BI | BC |
| W. | SFT Model | 79.1 | 62.0 | 94.2 |
| | +ORISE | 84.1 | 72.6 | 96.7 |
| M. | SFT Model | 75.4 | 76.3 | 94.3 |
| | +ORISE | 77.2 | 84.0 | 92.5 |

Table 5: Accuracy of conversational behaviors on unseen noises. "TT", "BI", and "BC" denote turn-taking, barge-in, and backchanneling,"W." and "M." indicate noise from Wham and MUSAN.

Figure 2: Visualization of $-\mathcal{R}_1$ (turns consistency) and $\mathcal{R}_2$ (backchanneling) during training.

## 6.3 Adaptation on Unseen Noises

In this section, we first analyze the impact of unseen noise on SDLM behaviors, and then examine the efficacy of ORISE to alleviate it. As shown in Table 5, "SFT Model" is trained with online augmented noisy segments from Wham training set, and we evaluate the model on Wham evaluation set and MUSAN noises. From the results, we observe that when noise is introduced during training, backchannel performance improves significantly (86.4%→94+%) comparing with result in Table 4, as the model becomes desensitized to third-party voices. However, the performance of both turn-taking and barge-in degrades notably under the same conditions. ORISE effectively mitigates these challenges without relying on labeled data, thereby demonstrating its capacity to preserve balanced conversational behaviors under domain shift.

## 7 Conclusion

In this paper, we introduce robust full-duplex SDLM enhanced by ORISE, a reinforcement learning method to improve model's conversational behaviors in real-time interaction. The proposed ORISE leverages automated annotations of online generated speech to guide the model toward well-formed and speech-text aligned responses. Through extensive experiments, our model demonstrates strong performance in handling user barge-in, turn-taking, and backchanneling, while remaining resilient to unseen background noise without the reliance of labeled data. These results highlight that our approach enables SDLMs to handle diverse interactive spoken dialogue dynamics within a unified system, thereby serving as a universal optimization framework for next-generation conversational AI.

## Ethics Statement

Our research utilizes publicly available and synthetic data, ensuring that no private or sensitive information is employed. We have carefully considered potential biases inherent in conversational models and have designed our approach to minimize harmful behaviors. The proposed system is intended for academic exploration and serves as a prototype for further research; its future deployment in real-world applications should be accompanied by comprehensive ethical review and oversight. We also encourage future studies to further investigate the societal and ethical implications of next-generation conversational AI systems.

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

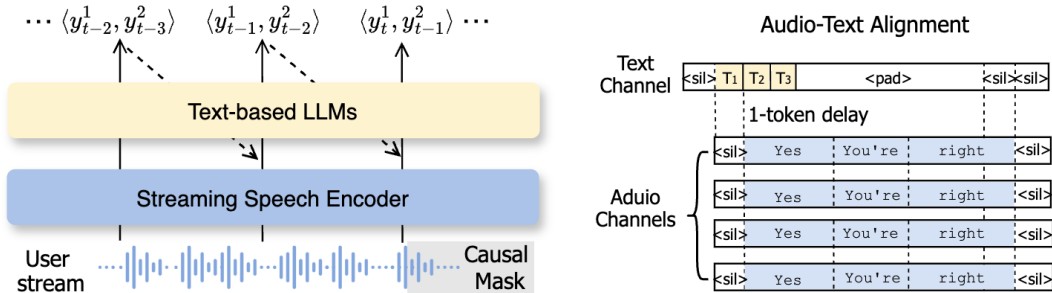

Figure 3: Full-duplex Model structure (left) and data alignment between audio and text channels (right). We visualize 1 turn of agent response to illustrate the predicted channels of LLM. "$T_1 \sim T_3$" are the text tokens and the blue chunks are composed by audio tokens.

# A  Appendix

## A.1  Details of Training Data

Figure 4 illustrates the format of our training data. Since the agent cannot anticipate the user's behavior in advance, we introduce an offset of $\Delta$=0.64 second into the training labels for both turn-taking and barge-in. This offset enables the agent to detect the appropriate timing from the user's speech input and exhibit the correct conversational behaviors. Each token in the figure corresponds to 80 ms in real-world time.

Statistics of our training data are presented in Table 6. For MS MARCO and Alpaca, we adopt their predefined test splits. For other synthetic datasets, we reserve an unseen shard (approximately 5 hours) as the test set.

Table 6: Synthetic training data with multi-turn and barge-in.

| Task | Dataset | #Hours | Speech | Multi-turn | Barge-in |
|------|---------|--------|--------|------------|----------|
| Spoken QA | ASR-QA | 20k | Mix | Augment | ✗ |
| | MS MARCO | 0.2k | TTS | Augment | ✗ |
| | Alpaca | 0.2k | TTS | Augment | ✗ |
| | Internal SFT | 3k | TTS | Real | ✓ |
| Conversation | UltraChat | 3k | TTS | Augment | ✓ |
| | Topic | 0.3k | TTS | Augment | ✓ |

## A.2  Preference-optimization Baseline

To enable preference optimization, we utilize the implicit reward modeling (Rafailov et al., 2024) though the log-probability between policy model $\pi_\theta$ and reference model $\pi_{\text{ref}}$:

$$\mathcal{R}(Y^1, Y^2, X) = \lambda_1 \log \frac{\pi_\theta(Y^1|X)}{\pi_{\text{ref}}(Y^1|X)} + \lambda_2 \log \frac{\pi_\theta(Y^2|X)}{\pi_{\text{ref}}(Y^2|X)} \tag{8}$$

where $\lambda_1$ and $\lambda_2$ keep consistent with SFT to balance the weight of channels. Given user input $X$, we utilize online sampling to obtain a batch of text-speech pairs $\mathcal{B} = \{\langle Y_i^1, Y_i^2 \rangle\}_{i=1}^B$. Using $\mathcal{C}_1$ and $\mathcal{C}_2$, we filter positive and negative samples from $\mathcal{B}$:

$$\mathcal{B}^+ = \{\langle Y_i^1, Y_i^2 \rangle \in \mathcal{B} \mid (\mathcal{C}_1 \wedge \mathcal{C}_2 \wedge \mathcal{C}_3)(\langle Y_i^1, Y_i^2 \rangle)\}, \tag{9}$$

$$\mathcal{B}^- = \{\langle Y_i^1, Y_i^2 \rangle \in \mathcal{B} \mid \neg(\mathcal{C}_1 \wedge \mathcal{C}_2 \wedge \mathcal{C}_3)(\langle Y_i^1, Y_i^2 \rangle)\}. \tag{10}$$

where $\wedge$ is the logical "and" and $\neg$ is logical "not". For DPO baseline (Rafailov et al., 2024), we reduce the $\beta$ for negative examples in its loss function, as we observed that in our system,

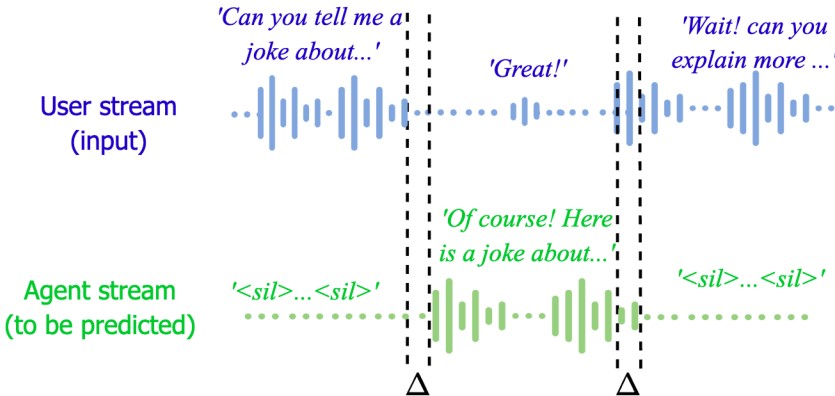

Figure 4: Multi-turn training data formulation, where Δ is set as 0.64 second for agent to take the turn and handle user bargi-in. The backchannel utterance (i.e., "great") is synthesized and insert into the training data points with > 4 silence.

positive and negative samples frequently yield identical predictions (i.e., the pad token). Maintaining the same $\beta$ for both positive and negative samples introduces high instability during training.

### A.3 Limitations and Future works

One limitation of this work is that our speech-to-speech modeling framework requires full fine-tuning of the LLM, which restricts our current exploration to relatively small models around 1.5B parameters. In future work, we plan to adopt larger LLMs as backbones to investigate whether they lead to substantial improvements in reasoning capabilities. Another important research direction is how to effectively preserve the real-world knowledge encoded in pretrained LLMs during SDLM modeling. As observed in challenging datasets such as UltraChat, there remains a performance gap between our system and the GT+LLM baseline as shown in Table 3. This highlights a promising avenue for future work aimed at better integrating knowledge retention with interactive spoken dialogue generation.

Furthermore, we find that evaluating dialogue data is inherently complex and challenging. Beyond the content generated by the agent, a wide range of paralinguistic cues—such as emotion, speaker identity, and interaction style—must also be considered for a intelligent AI system. These factors play a critical role in reinforcement learning, as they directly impact reward design and model behavior. In future work, we plan to develop a more comprehensive evaluation framework to support continued progress in this field.

