# OpenReview forum: "Reinforcement Learning Enhanced Full-Duplex Spoken Dialogue Language Models for Conversational Interactions"
_colmweb.org/COLM/2025/Conference — COLM 2025_

### Official Review · Reviewer_mDwK · 2025-05-06

**Rating:** 6
**Confidence:** 5
**Ethics Flag:** 1

**Summary:**

This paper addresses the challenge of enhancing full-duplex spoken dialogue language models by introducing an online reinforcement learning optimization method (ORISE) for more robust and natural conversational interactions. The main contributions include the development of ORISE, a reinforcement learning-based framework that utilizes automated annotations to refine conversational behaviors such as turn-taking, user barge-in, and backchanneling, alongside demonstrating robustness in noisy and unseen conditions. The paper compares this approach against relevant full-duplex baselines (e.g., Moshi, Freeze-Omni), effectively considering LLM baselines.

**Questions To Authors:**

1. Could you clarify in more detail how the automated annotation method reliably differentiates between subtle conversational behaviors like backchanneling and user barge-in, especially in noisy environments?
2. Have you considered or experimented with human-in-the-loop evaluations or human preference modeling to further validate your automated annotation-based reward scheme?
3. How sensitive is ORISE to hyperparameter selection, particularly regarding reward weights and KL divergence terms, and how was stability during training ensured?

**Reasons To Accept:**

1. Introduces a novel reinforcement learning-based optimization framework specifically tailored for improving conversational dynamics in full-duplex SDLMs.
2. Provides comprehensive experimental evaluation showing significant improvements in critical interaction metrics (e.g., user barge-in accuracy, turn-taking latency, backchannel handling).
3. Demonstrates the generalization capability of the proposed approach to unseen background noise conditions without reliance on supervised labels.

**Reasons To Reject:**

1. Limited clarity on how exactly automated annotations from the speech evaluations are guaranteed to consistently reflect desirable conversational qualities, potentially affecting reproducibility.
2. While noise robustness is demonstrated, the analysis on how varying noise conditions specifically impact conversational performance metrics remains somewhat superficial.
3. Lack of human evaluation or user studies to validate that improvements in metrics truly reflect a better user experience, potentially limiting real-world impact.

---

> ### Author Response · Authors · 2025-06-02
> **Response to Reviewer mDwK**
>
> We sincerely appreciate your thorough review and valuable comments. Your expertise has significantly contributed to improving this work. Below we provide detailed responses to address your concerns.
>
> **Question1: Introducing human preference annotation in the learning loop**\
> Thanks for your valuable question. We clarify that this work focuses on improving conversatioanl interaction accuracy (good/bad judgments) rather than subjective preference rankings. As evidenced by our listening examples: Human perception shows no significant preference between 0.4s vs 0.6s response latencies, while VAD models already provide precise timing judgments. We contend human annotation’s primary value lies in assessing response content quality, not microsecond-level timing nuances.
>
> **Question2: Reward Modeling**
> - We highlight our intentional avoidance of excessive hyperparameters in reward modeling to ensure reproducibility. The KL coefficient (0.2) is not a sensitive hyperparameter and aligns with standard RLHF practices, serving to prevent overly aggressive policy updates.
>
> - Due to conversational dynamic, reward components exhibit varying effectiveness across different optimization set. On UltraChat (Main Experiments), R1 (Turn Consistency) and R2 (Backchannel/Barge-in) dominate so that we present them in Figure2, while R3 (Audio-Text Alignment) shows minimal impact. However, on less-turn datasets (ASR-QA): R1 becomes negligible since all reponse predict correct turns; on other datasets ( like topic):  R2’s backchannel/barge-in rarely triggers with short agent responses. Therefore, our solution aggregates directly measurable rewards through simple summation. While preference-based optimization was explored to eliminate manual weight tuning, Table 4 demonstrates its inferior performance compared to our approach.

---

### Official Review · Reviewer_nQWA · 2025-05-13

**Rating:** 6
**Confidence:** 4
**Ethics Flag:** 1

**Summary:**

This paper presents ORISE (Online Reinforcement with Interactive Speech Evaluation), a reinforcement learning framework for enhancing full-duplex spoken dialogue language models (SDLMs). The authors focus on optimizing conversational behaviors such as turn-taking, user barge-in, and backchanneling through a customized reward function derived from automated annotations. The paper makes contributions in adapting text-based LLMs into full-duplex SDLMs and demonstrating how RL can improve conversational dynamics without relying on labeled data, even in unseen noise conditions.

The work represents a meaningful step toward more natural and responsive conversational AI systems that can better handle the complex dynamics of human conversation. The empirical results show improvements in both accuracy and latency for handling important conversational behaviors compared to baseline models.

**Questions To Authors:**

Some minor typos:

page 3, line 105, aim to optimized -> aim to optimize

page 5, line 206: are keep consistent -> are kept consistent

Also, I suggest you move Figure 3 and 4 to the main text to help the readers have a better view of the paper.

**Reasons To Accept:**

- **Technical Innovation**: The paper introduces a novel approach combining RL with automated speech evaluation to enhance conversational behaviors, which is valuable for advancing SDLMs.

- **Forward-looking Vision**: Full-duplex SDLMs represent an important frontier in conversational AI, and this work makes progress toward more natural human-computer interaction.

- **Efficient Architecture**: The authors demonstrate strong performance with a relatively lightweight model (1/6 the parameter size of comparable models), which is important for practical applications.

- **Adaptability**: The framework shows promising results in handling unseen noise conditions without needing labeled data, demonstrating practical utility for real-world deployment.

**Reasons To Reject:**

- **Data Limitations**: The training methodology relies heavily on synthetic data with artificially constructed turn-taking behaviors. The paper acknowledges the use of TTS models to synthesize conversations from text datasets and introduces artificial timing delays (such as Δ=0.64 second which seems arbitrary). This synthetic approach may not capture the natural variability and nuance of real human conversational dynamics.

- **Evaluation Methodology Concerns**: The reward mechanisms are primarily based on automated metrics from VAD models without sufficient human evaluation of conversational naturalness. This creates a risk of reward hacking where the model optimizes for the metrics rather than for genuinely improved conversation quality.

- **Reward Design Limitations**: The paper combines multiple rewards (R1, R2, R3) through simple summation without exploring how different weightings might affect model behavior. There's no ablation study or analysis showing how these weights interact or whether the chosen combination is optimal.

- **Limited Real-world Validation**: While the paper demonstrates improvements on automated metrics, there's insufficient evidence that these improvements translate to better subjective experiences in real human-computer interactions.

---

> ### Author Response · Authors · 2025-06-02
> **Response to Reviewer nQWA**
>
> We sincerely appreciate your thorough review and valuable comments. Your expertise has significantly contributed to improving this work. We would revise our paper according to your sugeestions and corrections, and we provide detailed responses to address your concerns.
>
> **Question1: Data Limitations**
> Thank you for your feedback. We wish to clarify two key points: (1) Reliance on Synthetic Data. Existing conversational datasets can NOT train a helpful spoken agent because neither speaker in these dialogues (e.g., Fisher Corpus conversations) fulfills the role of a qualified agent. This fundamental role mismatch necessitates synthetic data for effective agent training. (2) 0.64s Delay Rationale. The delay is essential because the agent cannot inherently preddict when a user will stop speaking or attempt to interrupt. Our experiments demonstrate that this empirically determined threshold, within reasonable bounds, does not disrupt natural interaction patterns. As evidenced by our provided listening examples, the agent exhibits natural latency behaviors in both turn-taking and barge-in scenarios.
>
> **Question2: Reward hacking and reward design.**\
> Thank you for your question. We would like to clarify that the VAD metric in reward modeling does not induce reward hacking. As shown in Table 3, the introduction of IORISE does not degrade response quality (measured by GPT score), while maximizing the reward indeed improves the agent's conversational performance in terms of interaction accuracy. Our reward design prioritizes simplicity to avoid introducing excessive hyperparameter. Experimental results demonstrate that simply summing the reward components achieves good performance.
>
>
> **Question3: Limited Real-world Validation.**\
> We have provided unseen samples (both **topic** and **backgournd noises**) on the demo page, including multiple stress tests (e.g., rapidly interrupting the agent multiple times within short intervals). The agent consistently demonstrates polite responses and accurate content in these scenarios. We believe this provides sufficient evidence to demonstrate our system's readiness for real-world deployment.

---

> > ### Comment · Reviewer_nQWA · 2025-06-09
> >
> > I appreciate the authors rebuttal. Here're my responses to these points:
> >
> > 1. The data limitations are indeed a problem for academic research and need further investigations for publicly available data. One key question to investigate is whether models trained with synthetic data proposed in this paper is transferable to real agent data collected from human sources (like professional voice agents recorded while interacting with clients). This type of data is super expensive to collect so the work would be a definite accept if the authors can demonstrate that models with synthetic data can be fine-tuned with only a few hours of data of real human agent data and works as well as ground truth.
> >
> > 2. The metric is still evaluated automatically without any human subjective evaluation, so objective evaluations alone are still vulnerable to reward hacking,
> >
> > 3. This has addressed my concern regarding point 3, but my concerns from previous two points remain.

---

### Official Review · Reviewer_W4Zs · 2025-05-13

**Rating:** 6
**Confidence:** 3
**Ethics Flag:** 1

**Summary:**

The paper applies the reinforce algorithm to a duplex (speech in, speech out) conversational dialogue system. The authors show that after defining a set of rewards (which do not require human labels but are based on table stakes properties of the conversation which can be measured automatically with existing systems like VAD) and a means of combining these, that the resulting system is better at turn taking, handling barge ins and backchannels.

**Questions To Authors:**

- Regarding C-3 and the alignment between text and speech: if perfect alignment (as in words are the same) is desired, is a speech system needed? Can an independent TTS system not be used in such a case?
- in equation 2: why the need for 2 hyperparameters? Isn't one sufficient?
- S2S section (starting line 159): why is the conversational data insufficient? Not pushing back on the claim, but asking for a brief explanation/reason why it's insufficient.
- Further in the same paragraph, around line 162: why does silence need to be inserted?
- nit: there's a typo "an" -> "a" on line 160 too.

**Reasons To Accept:**

The paper is clearly written and shows improved results on barge ins on 2 public datasets. Results are mixed on a QA task, with the proposed method improving upon the SFT starting point in some cases and regressing on others.
I don't know the current literature in this space very well, but I believe there is enough in this paper that it will be of interest to the multimodal LLM community.

**Reasons To Reject:**

No significant reasons. This is the application of existing techniques to a (to my knowledge) new area.
The results as noted are mixed. There does appear to be a trade off across the turnTaking, bargeIn and backChannel properties that are aimed to be optimised here, but overall the proposed rewards and optimisation of them via reinforce does lead to benefits in most cases, particularly with noise introduced into the speech (table 5).

---

> ### Author Response · Authors · 2025-06-02
> **Response to Reviewer W4Zs**
>
> We sincerely appreciate your thorough review and valuable comments. Your expertise has significantly contributed to improving this work. Below we provide detailed responses to address your concerns.
>
> **Question1: Can an independent TTS system not be used in text-speech alignment** \
> Thank you for your question. Two critical factors prevent the use of independent TTS systems in this context: (1) Real-time Requirements. Our real-time duplex design requires **token-level streaming** speech synthesis—each LLM prediction must generate 80 ms of synthesized speech including silence. This ensures a theoretical system latency of 80 ms, enabling rapid handling of dynamic conversational scenarios. Existing mainstream TTS degins are non-causal, and even causal TTS implementations hardly achieve such low systemic latency. (2) Input Differences. Conventional TTS systems rely on text or semantic tokens for pretraining. In contrast, our system performs audio token prediction directly based on the LLM's hidden representations. The LLM implicitly learns an integrated TTS system while simultaneously predicting text, enabling concurrent "thinking" and "speaking" during responses.
>
> **Question2: Why exisiting conversational data is insufficient to train a helpful agent**\
> We appreciate the opportunity to clarify this important distinction. We summarize two reasons: (1) Speaker Role Mismatch in Conversational Datasets. Datasets like Fisher Corpus contain dialogues between peers (e.g., friends), where **neither spaker assumes an agent role** capable of providing structured assistance. Training on such data produces generic conversational models rather than goal-oriented agents specialized in delivering helpful recommendations. (2) The speaker voice in conversational datasets is diverse while we just need a fixed voice for agent. Though zero-shot TTS techniques could theoretically enable diverse vocal personas, our current priority focuses on establishing core competency in polite and effective duplex interaction.
>
> **Question2: Why does silence need to be inserted?**\
> Unlike turn-based models, the full-duplex model operates as a real-time system in continuous prediction mode, as illustrated in Figure 1. Specifically, When the user is speaking, the agent must **remain silent** (i.e., predict silence tokens) to actively listen. Consequently, silence segments must be explicitly inserted into training data to teach the agent when to withhold speech production.
>
> **Question3: Why the need for 2 hyperparameters in EQ2**\
> Thank you for your suggestion. A single parameter is indeed sufficient. We initially included two parameters in the formula to explicitly illustrate the one-to-one correspondence between weights and channels. We will revise this along with your other recommendations in the next version.
>
> We hope these responses have addressed your concerns, and we reiterate our gratitude for your efforts in reviewing this work.

---

### Official Review · Reviewer_PVVb · 2025-05-17

**Rating:** 6
**Confidence:** 4
**Ethics Flag:** 1

**Summary:**

This paper presents a novel method for better training full-duplex spoken dialogue language models (SDLMs) with automatically generated supervision. Specifically, the model consumes input audio, previous text outputs, and audio outputs (as tokens), and produces the next text tokens and audio output (as tokens). To better handle complex scenarios where the user interrupts the AI assistant (“barge-in”), utter affirmative sounds (”backchanneling”), this paper proposes a reward modeling approach that leverages pretrained ASR and VAD models to compute the rewards of the model outputs. Then, they utilize reinforcement learning to maximize the reward. Empirical results show that their method outperforms existing full-duplex SDLMs in terms of latency and prediction accuracy for barge-in and backchanneling situations.

**Questions To Authors:**

- Line 105: “optimized” → “optimize”.
- Table 4: The BLEU score seems very low, and applying RL further reduces it. Is this normal?
- It seems that the reward modeling in ORISE does not take into account the audio-specific features, such as emotions. Is this true?
- Are there other kinds of backchanneling other than affirmative short utterances?
- In Algorithm 1, I think the ASR model required for computing $R_3$ should be included.
- Line 298: System ID-2, 3, and 4 have not been introduced, and I have a hard time understanding what they refer to.
- Figure 2: I think it’s better to report $R_1$ instead of $-R_1$.
- Table 5: I suggest including the score for the model without unseen noise for reference.

**Reasons To Accept:**

- The paper address an important research question and the empirical results are great.
- The experiments clearly validates the effectiveness of the proposed model.
- The paper is generally easy to follow and the motivation is clear.

**Reasons To Reject:**

- There are three reward functions, but there is no ablation studies to verify the effectiveness of each of them. Thus, it is hard to know the effectiveness/necessity of each of the rewards.
- While I understand that the text outputs are not the focus of this paper, I think providing the evaluation of the output text quality is very important for understanding the implications of this paper. I suggest, at the very least, to include such results in the Appendix.
- It seems that the model requires an ASR model to produce the text labels for the audio, which is a kind of supervision. Does this align well with the statement in the Introduction that ORISE do not rely on any supervised labels?
- To better understand the results, I suggest including the performance of non-full-duplex SDLMs.

---

> ### Author Response · Authors · 2025-06-02
> **Response to Reviewe PVVb**
>
> We sincerely appreciate your thorough review and valuable comments. Your expertise has significantly contributed to improving this work. Below we provide detailed responses to address your concerns.
>
> **Question1: Clarification on ASR Model**\
> We would like to clarify a misunderstanding here: When calculating the Word Error Rate (WER) by an ASR model, we measure the discrepancy between the **predicted speech** and **predicted text** (our duplex model predicts them at the same time), rather than comparing generated speech with ground-truth labels. This process does not rely on any external supervision. Like the VAD model, it serves solely as an unsupervised reward evaluator.
>
> **Question2: Low BLEU Score in Table 4**\
> On the UltraChat dataset, a BLEU score of 10 is normal. This dataset was adapted from the text-based UltraChat (https://github.com/thunlp/UltraChat) into a conversational format, with intentionally open-ended and diverse topics. As shown, even text-based GPT responses have a human evaluation score of only 6.4, while our 1B model achieves a GPT evaluation score of 4.0, surpassing the 7B Moshi model's 3.4.
>
> **Question3: Emotion in ORISE reward modeling**\
> Yes, emotion is not modeled in current system. To ensure stability in agent speech synthesis, we intentionally used neutral emotional profiles during data synthesis, resulting in stable agent speech generation. However, your insight raises an intriguing point: our subsequent experiments reveal that when provided with emotionally varied speech labels for training, the current codec system can indeed generate agent speech with richer emotional expressions. We would explore this direction in future work.
>
> **Question4: Backchanneling type**\
> The current user backchanneling only includes 1-5-word affirmative utterances. If the utterances become too long, the agent would struggle to determine whether the user is providing affirmation or attempting to initiate a new conversation.
>
> **Question5: Inluustration of System ID-2, 3, and 4**\
> Table 3 demonstrates our duplex system's capability to function as a real-time agent delivering accurate responses. The GPT score serves as the primary quality metric for answers, while conversational metrics are **excluded** since they measure response etiquette in coversation. Key Comparisons: (1) ID-2 vs ID-3: Both models were trained with direct SFT on our dataset. This comparison confirms that our duplex system can effectively adapt different text-based LLMs into duplex-capable agents. (2) ID-3 vs ID-4: Since the ORISE-enhanced ID-4 model shows significant improvements in conversational performance metrics in Table 2, in this Table 3, we need to confirm that it still maintains comparable GPT scores, proving that ORISE optimizations preserve answer quality.
>
> **Question6: including the score for the model without unseen noise for reference.**\
> Thank you for your suggestion. We wish to clarify a potential misunderstanding: The SFT model in Table 5 was the models trained without unseen noise. However, we note that if a model is trained entirely without noise and encounters sudden noise in the test set, the agent would fail to take the turn, as it would misinterpret noise as the user maintaining a keep-speaking mode. Thus, such a reference lacks meaningful comparative value. To make sure the delpoyment in reality, noise-augmentation is neccessary for dupelx training.
>
> We will incorporate your formatting and typographical suggestions in the next version. Thank you once again for your thorough review efforts.

---

> > ### Comment · Reviewer_PVVb · 2025-06-04
> >
> > I greatly appreciate your response.
> >
> > There are still some unaddressed concerns, such as:
> >
> > - Can you provide some ablation studies involving the three reward functions?
> > - Can you provide the results on non-duplex models as a reference (not a baseline)?
> > - Can you provide some evaluations of the output text quality of the model?

---

### Decision · Program_Chairs · 2025-07-08

**Decision:**

Accept

**Comment:**

This paper presents a new RL method to enhance full-duplex spoken dialogue language models.

All reviewers appreciate the contributions of this paper including important task, experimental results, and clear writing with marginal acceptance scores.

AC also agrees with the reviewers' consensus, so recommends accepting this paper.

AC asks the authors reflect the reviewers' comments in the final version.